# The impact of perinatal maternal stress on the maternal and infant gut and human milk microbiomes: A scoping review

**Niamh Ryan**[ID][1]*, **Siobhain O'Mahony**[2], **Patricia Leahy-Warren**[1], **Lloyd Philpott**[1], **Helen Mulcahy**[ID][1]

**1** School of Nursing and Midwifery, University College Cork, Cork, Ireland, **2** Department of Anatomy and Neuroscience, APC Microbiome Ireland, University College Cork, Cork, Ireland

* 122117696@umail.ucc.ie

## Abstract

### Background

Perinatal maternal stress, which includes both psychological and physiological stress experienced by healthy women during pregnancy and the postpartum period, is becoming increasingly prevalent. Infant early exposure to adverse environments such as perinatal stress has been shown to increase the long-term risk to metabolic, immunologic and neurobehavioral disorders. Evidence suggests that the human microbiome facilitates the transmission of maternal factors to infants via the vaginal, gut, and human milk microbiomes. The colonization of aberrant microorganisms in the mother's microbiome, influenced by the microbiome-brain-gut axis, may be transferred to infants during a critical early developmental period. This transfer may predispose infants to a more inflammatory-prone microbiome which is associated with dysregulated metabolic process leading to adverse health outcomes. Given the prevalence and potential impact of perinatal stress on maternal and infant health, with no systematic mapping or review of the data to date, the aim of this scoping review is to gather evidence on the relationship between perinatal maternal stress, and the human milk, maternal, and infant gut microbiomes.

### Methods

This is an exploratory mapping scoping review, guided by the Joanna Briggs Institute's methodology along with use of the Prisma Scr reporting guideline. A comprehensive search was conducted using the following databases, CINAHL Complete; MEDLINE; PsycINFO, Web of Science and Scopus with a protocol registered with Open Science Framework DOI 10.17605/OSF.IO/5SRMV.

### Results

After screening 1145 papers there were 7 paper that met the inclusion criteria. Statistically significant associations were found in five of the studies which identify higher abundance of potentially pathogenic bacteria such as Erwinia, Serratia, T mayombie, Bacteroides with higher maternal stress, and lower levels of stress linked to potentially beneficial bacteria

**Data availability statement:** All relevant data are within the article and its supporting information files.

**Funding:** The author(s) received no specific funding for this work.

**Competing interests:** No authors have competing interests.

such Lactococcus, Lactobacillus, Akkermansia. However, one study presents conflicting results where it was reported that higher maternal stress was linked to the prevalence of more beneficial bacteria.

## Conclusion

This review suggests that maternal stress does have an impact on the alteration of abundance and diversity of influential bacteria in the gut microbiome, however, it can affect colonisation in different ways. These bacterial changes have the capacity to influence long term health and disease. The review analyses data collection tools and methods, offers potential reasons for these findings as well as suggestions for future research.

## Introduction

Maternal health refers to the wellbeing of women during pregnancy, childbirth and the post-partum period [1]. The importance of maternal health is multifaceted as it impacts not only the mothers immediate health but also influences the health and development of children. Pregnancy and the post-partum period are viewed as a period of transition and adaptation in which healthy women may be sensitive to the impact of psychosocial stress [2]. Such stress is reported to result from poor social supports, work/family responsibilities, pregnancy, and birthing complications [3–5]. Studies across the globe report a prevalence of perinatal maternal stress ranging from 5% to 93% with significant increases reported since Covid-19 [3,5–9].

During this perinatal period, from conception to one year old, healthy mothers are more likely to have healthy babies, as infants undergo vital growth and development with the formation of their metabolic, endocrine, neural, and immune systems [10]. Children exposed to perinatal maternal mental health disorders often experience adverse birth outcomes, including preterm birth, and low birth weight [11,12]. Moreover, researchers are expanding upon Barker's theory of the Developmental Origins of Health and Disease (DOHaD) to demonstrate that exposure to environments such as stress during the perinatal period can contribute to long-term risks to adverse infant outcomes that extend into adulthood. These include metabolic diseases like obesity, cardiovascular disease, diabetes, along with immunological disease and neurobehavioral disorders [4,13–29]. Obesity, high blood pressure, and high blood sugar rank among the top four risk factors for death and disease in developed countries [30], with the US estimating costs of such chronic conditions at $4.1 trillion in 2022 [31]. Maternal health therefore is vital for achieving Sustainable Development Goal (SDG) target 3.4, of reducing premature mortality from non-communicable diseases by one-third by 2030, and contributing significantly to Goal 3 of ensuring healthy lives and well-being for all [32]. Maternal perinatal stress represents a prevalent and modifiable factor that can significantly impact maternal health, as well as foetal and child development. Therefore, it warrants focused attention in both research and preventative healthcare efforts.

Over the past two decades, researchers have studied the mechanism of foetal programming on health and disease. In the last decade, there has been a specific focus on how perinatal mood disorders, including maternal stress, may transmit risk to infants, with building evidence of the involvement of the gut microbiome in this process [24,33,34]. The gut microbiome is made up of trillions of bacteria, fungi, and other microbes, it plays an essential role in health and is involved in priming and maintaining gut, immune and metabolic health [35–37]. Psychological stress can influence the maternal gut and human milk microbiome structure via the microbiome-brain-gut axis, a communication pathway allowing microbes and their products to impact brain function and vice versa. It has been almost a decade since data from

animal models have linked prenatal stress exposure to changes in the hypothalamic-pituitary-adrenal axis (HPA), brain-gut axis functioning, and gut dysbiosis or imbalance in the offspring [38–41]. These alterations had lasting effects on metabolism, cognition, and behaviour in mice [38,39,42–44]. Recent evidence identifies that the immature human gut microbiome is modulated by many influencing factors such as maternal health, human milk, mode of delivery, antibiotics [45–50]. The data shows that these variables can change the mother's microbiome community, which can then transfer an abnormal microbiome community to the infant, predisposing them to an obesogenic, pro-inflammatory gut microbiome. Changes in microbial communities also focus on alpha diversity referring to the richness of the bacteria in the microbiome, and beta diversity which looks at the amount of differentiation within the bacterial community [51]. This transmission of faecal, vaginal and human milk microbiota from mother to infant is a process known in the literature as vertical programming [52].

Human milk has been referred to as "mother natures prototyped probiotic food" (McGuire and McGuire, 2015.p.g 1), with the human milk oligosaccharides (HMOs) in human milk promoting the growth of probiotic type bacteria such as Bifidobacterium [53]. Disputing previous theories that human milk was sterile, the evidence reports that the human milk microbiome is populated with bacteria, fungi, virus, and yeast essential for the present and future health of an infant [54–56]. However, recent evidence identifies the human milk microbiome is susceptible to maternal factors of body mass index, diet, mode of delivery, general health, and lactation stage, that can also influence the microbial seeding of an infant [45,57–59]. Despite this, much of the research to date has focused on the influencing factors on many other components of human milk such as cortisol, immune factors, and macro nutrients in human milk [60–67], with only a handful of studies focused on the milk microbiome [55,56,59,68,69].

Recent research has advanced understanding of the gut microbiome's role in health and disease, including factors influencing maternal and infant gut microbiomes and the human milk microbiome. However, limited studies examine the impact of maternal perinatal stress on these microbiomes. Initial screening reveals less than ten experimental studies over nine years resulting in overall mixed findings [70]. To date, there has been no systematic effort to map or review evidence within these concepts, therefore a scoping review is timely and warranted. Arksey and O'Malley's seminal work describes a scoping review as valuable for exploring the extent, range, and nature of research, as well as identifying gaps in the evidence base [71] The review aims to address the following question: What is known about the impact of perinatal maternal stress on the maternal and infant gut and human milk microbiomes? The objectives are to identify a) the methods and tools used by which stress and the microbiome(s) are measured; b) the impact of perinatal stress on specific taxa in the microbiome(s): c) the impact of perinatal stress on microbiome(s) diversity; d) any gaps in the evidence.

## Materials and methods

### Study design

This scoping review was conducted following the guidelines for scoping reviews described by the JBI Manual for Evidence Synthesis and Preferred Reporting Items for Systematic Reviews and Meta-analyses extension for Scoping review (PRISMA-ScR) [72,73]. The 27 item checklist was incorporated throughout the review and detailed in (S1 File 1). The protocol for this scoping review is registered with open science framework DOI 10.17605/OSF.IO/5SRMV and published with PLOS one following a peer review process [74].

### Search strategy

A three-step search strategy process was followed as recommended for JBI reviews. An initial limited search of PubMed and CINAHL was undertaken to analyse the text words, keywords

and index terms contained in the titles and abstracts. This informed the development of a second search strategy using all identified keywords and index terms across all included databases. Synonyms and interchangeable terms for these keywords were also identified. The keywords and search terms were peer reviewed and approved by a university librarian using the Peer Review of Electronic Search Strategy guidance (PRESS) [75] (S1 File 2). Thirdly, the reference list of all identified articles was searched for additional studies.

Multiple top up searches in the literature were completed within each research concept to broaden the results within the concepts (See S1 Table 1). The search strategy included the Boolean terms "OR"/"AND," Medical Subject Headings (MeSH), CINAHL headings and truncation "*". Varied combinations of search terms and MESH terms that were unique to each database were used in this search strategy. The following databases and search engines were included: CINAHL Complete (EBSCO), Psych Info (EBSCO), Pubmed, Web of Science and Scopus. The final date of all databases searched was 30th May 2024. There was a restriction of the final analysis to articles published in English as a matter of convenience and to avoid interpretative errors in attempted translation. We recognize the potential for language bias; therefore, the initial search included articles written in any language to quantify the extent of non-English literature excluded from our analysis and there were none.

## Study selection: eligibility criteria

The inclusion and exclusion criteria matched the population, concepts and context framework recommended by the Johanna Briggs Institute (JBI) can be viewed in Table 1 [72].

**Population.** Mothers and infants are the focus of the population as there is a substantial body of evidence indicating that maternal factors are influencing a transmission of risk to the infant in adverse conditions[4,21,23,56,76]. Infants older than 6 months and those formula fed were excluded as both factors can bring their own changes in the microbiome [77].

**Table 1. Inclusion and Exclusion Criteria.**

| | | Inclusion Criteria | Exclusion Criteria |
|---|---|---|---|
| **Population:** | **Mothers** | • Mothers from conception up to 6 months post-natal | • Mothers who are more than 6 months post-natal. |
| | **Infants** | • Infants from birth to 6 months old who are exclusively breastfeeding. | • Infants older than 6 months. Preterm infants<br>• Infants born with birth defects/anomalies<br>• Infants who combination feed or formula feed. |
| **Concepts:** | **Maternal stress** | • Studies reported on maternal stress identified by blood sampling or self-assessment tools.<br>• Studies reporting on maternal stress and anxiety will be included if data on stress can be extrapolated as a single variable. | • Studies reporting on other mental health disorders such as depression, anxiety, bipolar, psychosis. |
| | **Gut Microbiome** | • Studies reporting on gut microbiome as key outcome of mothers or infants in the above ages with laboratory analysis and sequencing of faecal microbiome. | • Studies focused on neurodevelopmental/ neurocognitive or allergic diseases as key outcomes along with gut microbiome.<br>• Studies evaluating probiotic treatments.<br>• Non-Human studies |
| | **Human milk microbiome** | • Studies focused on human milk microbiome (HMM) of the breastfeeding mother. Studies including discussion on all composition of breast milk included if data can be extracted regarding the human milk microbiome. | • Studies conducted with animals or non-human subjects.<br>• Studies evaluating probiotic treatments.<br>• Studies focused on other components of Human milk with no reference to the Human milk microbiome. |
| **Context** | **Perinatal period** | • Studies focus on mother from conception to 6 months<br>• Studies focusing on babies up to 6 months old who are exclusively breastfeeding. | • Studies that include maternal or infant data outside this specific perinatal period |

**Key concepts.** The review focuses on maternal stress and its impact on the maternal and infant gut and/or human milk microbiome. It examines the growing evidence of how exposure to stressful environments affects these microbiomes [25,78–80].

**Context.** The context is in perinatal period as much work has been done on the importance of the foundation of health in the first 1000 days from conception to two years where neural, immune and metabolic function are laying foundations for infant health [10,81–83]. The evidence suggests this is the most critical window for future health. Fig 1 outlines the Concept map of the PCC Framework for this study.

This review includes primary research with quantitative, qualitative and mixed methods studies. Secondary analyses, like systematic reviews, were considered if they met the other inclusion/exclusion criteria however none were found. Case reports, book chapters, guidelines, commentaries, editorials, letters to editors, and narrative reviews are excluded as these sources do not directly address the review question and pose challenges in data extraction. Limits were applied for papers from 2014 to present and those with English language.

Grey literature lacks rigorous peer review along with risk of bias and reliability concerns [84]. In the absence of ample published research, excluding grey literature ensures higher quality, minimizes bias, and maintains objectivity in the review process. The below Table 2 identifies final Eligibility Criteria for types of evidence to be included
Following the five database searches, all identified citations were collated and imported into Covidence with duplicates removed. Five primary reviewers (NR, HM, LP, SOM, PLW) independently screened all papers (n = 1154) in 3 stages: title screening, abstract screening,

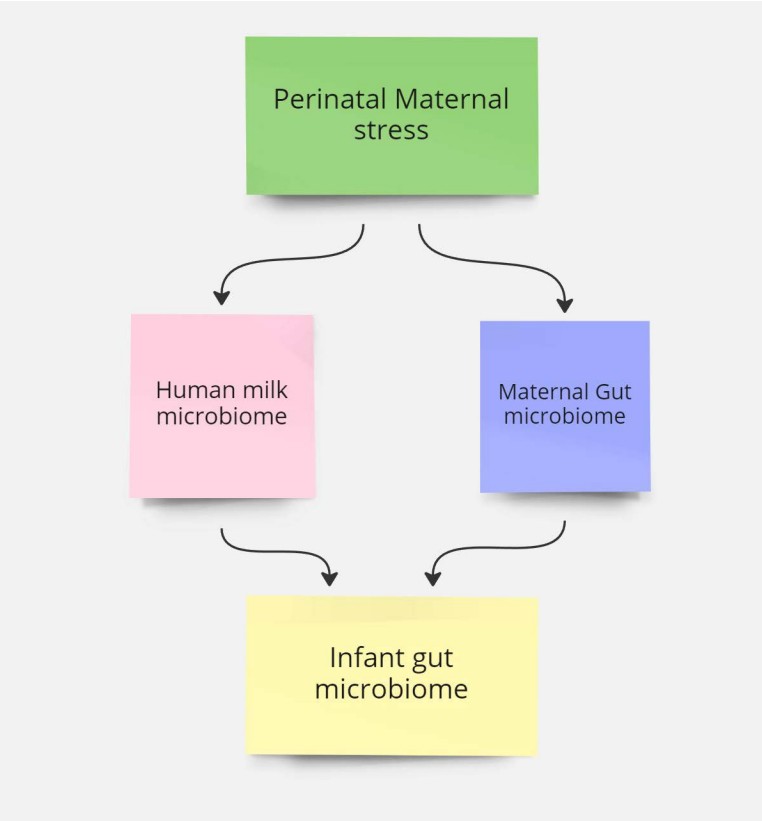

**Fig 1. Concept Map underlining the PCC.**

**Table 2. Final Eligibility Criteria.**

| Included | Excluded |
|---|---|
| **Primary Research:**<br>**Experimental and Observational Studies:**<br>• Quasi- experimental<br>• Cohort studies<br>• Case control<br>• Cross sectional<br>• Case series<br>**Qualitative studies** | |
| **Secondary research**:<br>• Systematic reviews and meta- analysis | **Secondary Research:**<br>• Narrative reviews |
| **Years of Publication** 2013–2024 | Before 2013 |
| | Grey literature or unpublished literature including case reports, book chapters, guidelines, commentaries, editorials, letters to the editors. |
| English | Non English |

and full-text screening. Conflicts at any stage were discussed with 4 primary reviewers NR HM SOM LP at online meeting with agreement by all required to exclude at all stages. The full text of selected citations (n = 35) was assessed in detail against the inclusion criteria by NR HM SOM LP. Reasons for exclusion of sources of evidence at full text that did meet the inclusion criteria was recorded. Exclusion at the full text stage required the agreement of 5 primary researchers NR HM LP SOM PLW at an online meeting and resulted in 7 studies. The results of the search are presented in a Preferred Reporting Items for Systematic Reviews and Meta-Analyses (PRISMA) flow diagram [72] (Fig 2).

## Data extraction

Data were extracted from the remaining 7 papers included in the scoping review by two independent reviewers for each study from the team NR HM LP SOM PLW, using a data extraction tool developed by the reviewer NR on Covidence (S1 File 3). Expert's recommend the review team to trial the extraction form on two or three sources to ensure all relevant results are extracted and to improve transparency and consistency [71,85–87], therefore a pilot test was completed by a second reviewer HM with no issues noted and allowed extraction to proceed. There was a separate meeting with two of the team NR SOM to ensure all statistical tests were correctly documented and key findings displayed for the 7 papers.

## Results

The electronic search strategy yielded a total of 1154 papers with 7 papers remaining post full text screening with inclusion and exclusion criteria applied in the population, concepts and context. The 7 papers aligned to the concept of perinatal stress and the maternal and infant gut microbiome only. Results from key findings from extracted data are presented in Table 3.

### Study characteristics

The 7 studies came from various global locations, there were three completed in the US [70,88], two in the Netherlands [89,90] with one in Finland [91], and one in Ecuador in South America [92]. Studies had wide ranging sample sizes from 25 participants to 446. Six of the seven studies had a sample size of less than 100 with a study by Aatsinki [91] having the largest sample of 446. The publication years ranged from 2015–2023. Two of the studies aimed

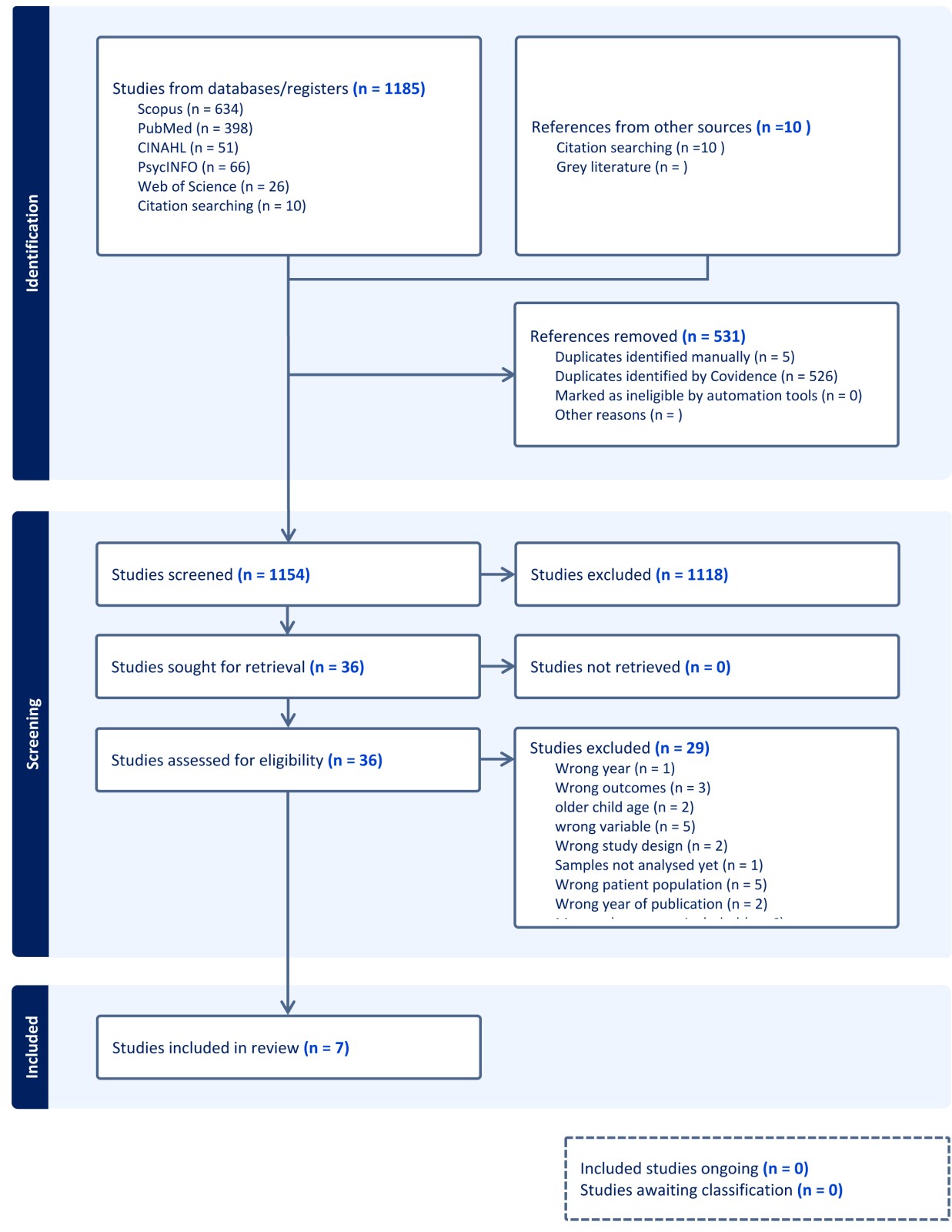

**Fig 2. PRISMA flowchart.**

**Table 3. Study Characteristics.**

| First Author Year Country | Aim: | Study Design Sample size = n | Variables/ Concepts in the Study PCC | Data Collection | | | | Co- variates included (not part main experiment but influence result) | Data Analysis | Key Findings: |
|---|---|---|---|---|---|---|---|---|---|---|
| | | | | Stress: Self-assessment tool: Timeline | Faecal sample test, Taxonomy Timeline | Objective Stress measure | | | Adjustment for Confounders Statistical methods | |
| Weiss S.J., & Hamidi M. 2023 United States of America (USA) | To assess the relationship between maternal stress during pregnancy and the diversity and composition of the neonate gut microbiome at one month of age. | Cross sectional study n = 51 | Maternal stress and infant gut microbiome | Perceived Stress scale (PSS) @31–41 weeks gestation | 16S Phylum Genus Microbial Diversity @ 1 month of age | No | **Maternal:** Maternal treatment with antenatal corticosteroids, obstetric risk, mode of delivery (vaginal vs caesarean), **Infant:** Gestational age, Neonatal morbidity, | Gestational age and neonatal morbidity were adjusted as significant relationships were shown in preliminary analysis Negative binomial linear models. ANOVA | Greater abundance in infants exposed to more stress - Lactobacillus (OTU_28; 11.55 effect size (ES), p < .001), Lactococcus (OTU_53; 8.88 (ES), p < .01) and Bifidobacterium (OTU_313; 6.62 (ES), p < .01) Significantly higher in abundance (p < .01) for the infants whose mothers were less stressed: Erysipelotrichaceae (OTU_12; 25.62(ES), p < .001).Eggerthella (OTU_43; 23.74 (ES), p < .001), Bacteroides (OTU_134; 22.57 (ES), p < .001), Bacteroides (OTU_18; 22.46 (ES).p < .001), and Ruminococcus (OTU_14; 9.66 (ES), p < .01).Enterobacteriaceae (OTU_298; 2.63 (ES), p < .05) and Enterococcus (OTU_262; 3.66, p < .05 Prenatal stress was significantly associated with greater alpha diversity (Simpson Diversity index(p = 0.02) |
| Peñalver et al., 2023 USA | To understand the maternal gut-microbiota-brain axis during the second and third trimesters in relation to Perceived Stress through integration of the maternal gut microbiota composition and maternal cytokine and chemokine concentrations in serum. | Cross sectional n = 84 | *Maternal stress and the maternal gut microbiome* | Reduced Perceived Stress scale (PSS- 10) Trimester 2 and 3. | 16S Species level Unclear | Serum Cytokines | Education. Civil status Age Region BMI | Generalized linear models adjusted for age, BMI Chi square T -tests Spearman's partial and non-partial correlations Perm nova | Perceived stress (PS) No significant differences in taxa alpha and beta diversity and volatility. PS positively associated with species Bacteroides uniformis (p < 0.05) and Terrisporobacter mayombei (p < 0.01) (Pathogenic) Higher PS had lower abundance of munciniphila (probiotic) Correlation co-efficient r2 = -4. Bacteroides negatively associated with CXCL11 and positively with PSS connecting immune factor and self-report measures. |

*(Continued)*

Table 3. (Continued)

| First Author Year Country | Aim: | Study Design Sample size = n | Variables/ Concepts in the Study PCC | Data Collection | | | | Co-variates included (not part main experiment but influence result) | Data Analysis | Key Findings: |
|---|---|---|---|---|---|---|---|---|---|---|
| | | | | Stress: Self-assessment tool: Timeline | Faecal sample test, Taxonomy Timeline | Objective Stress measure | | | Adjustment for Confounders Statistical methods | |
| Jahnke et al. 2021 Ecuador | To test the relationships among maternal precarity and HPA axis dysregulation during the peripartum period, infant gut microbiome composition, and infant HPA axis functioning | Cross sectional n = 25 | Maternal stress and the infant gut microbiome | Perceived stress scale (PSS) Pre partum, Postpartum 1 month and 2 months | 16S Phylum Genus Microbial Diversity @ 2 months | Salivary Cortisol Mother and infant HPA axis via morning cortisol/ cortisol Awakening-response (CAR) | **Maternal:** General socio-demographic Household size, Employment, Parity, Health behaviours, Education **Infant:** Mode of delivery, Infant feeding | Ran linear regression models and controlled for confounders with mode of delivery and infant feeding. Observation only of taxa. Mann Whitney U test with diversity measurement and stress Bivariate analysis | | Post-partum stress significantly associated with lower alpha diversity in infant stool at 2 month (p = 0.01) Stress in pregnancy sig ass with Clostridiaceae Clostridium (p = .02), Stress pp 2 moths Clostridiaceae Clostridium (p = <.01) however lost significance when adjusted with confounders High infant basal cortisol 3 days PP associated with lower Actinobacteria (p < 0.01), Bifidobacterium (P0.01) and higher Enterobacteriacae (p = 0.03) Low maternal CAR prepartum associated with lower abundance of Bacteroides (p = 0.01), post portum low CAR higher Veillonella (p=0.01) |
| Aatsinki et al. 2020 Finland, Europe | To determine the associations between chronic maternal prenatal stress and infant faecal microbiota composition and diversity parameters in a large prospective human cohort | Cross sectional n = 446 | Maternal stress and the infant gut microbiome | Daily hassles negative subscale @14, 24, 34 weeks gestation | 16S Phylum Genus Diversity @2.5 months of age | Hair cortisol conc. | **Maternal:** Age, education, SSRI/SNRI use, pre-pregnancy BMI, gestational week, **Infant:** birth weight, mode of delivery, categorical breastfeeding, infant antibiotic course, infant age at sampling | Linear and Graded adjustments for covariates - i.e, if statistically significant with descriptive statistics. Spearman's correlation. Perm nova - Bray Curtis. | | Maternal chronic PPD (all symptom measures) showed positive associations (FDR < 0.01) with bacterial genera from phylum Proteobacteria. Daily hassles neg subscale has a positive association with genera Erwinia, Serratia, haemophilus (F < 0.01). (Potentially pathogenic bacteria) None of the results of PPD questionnaire were associated with microbiota or diversity (FDR > 0.97) Low exposure to stress linked to abundance health promoting bacteria such as Akkermansia. (Effect size – 10) Increased cortisol related to decreased Lactobacillus (FDR <0.01) |

*(Continued)*

**Table 3.** (Continued)

| First Author Year Country | Aim: | Study Design Sample size = n | Variables/ Concepts in the Study PCC | Data Collection | | Co-variates included (not part main experiment but influence result) | Data Analysis | Key Findings: |
|---|---|---|---|---|---|---|---|---|
| | | | | Stress: Self-assessment tool: Timeline | Faecal sample test, Taxonomy Timeline | | Adjustment for Confounders Statistical methods | |
| | | | | | | Objective Stress measure | | |
| Hechler et al. 2019 Netherlands | Investigates the relationship between psychosocial stress and faecal microbiota in pregnant mothers | Cross-Sectional n = 70 | *Maternal stress and maternal gut microbiome* | Everyday Problem checklist (R = .88*) @33.9 weeks. (mean) | 16S Phylum Genus Microbial Diversity @33.9 weeks (m) | No | **Maternal:** Educational background, Parity, Civil status | No adjustment for confounders Significant Multivariate Correlation criterion. Pearson's correlations. Two tailed t-tests Principal Coordinate Analysis (PCoA) Principal Component Analysis (PCA) | *Associations were found with general anxiety and maternal microbiota at genus level (r = .71, p = 0.04) Lower anxiety – high abundance Eubacterium and Oscillopsia. P <0.05* No significant associations were found with maternal stress variable and phylum level microbiota (p=1.0) No associations with general or pregnancy related stress at genus level. (p = 1.0) NO specific patterns or changes to diversity (p = 0.92 and p = 0.8) |
| Hu et al. 2019 US | To determine whether maternal anxiety, depression and stress during pregnancy is associated the bacterial diversity of the infant meconium | Cross sectional n = 75 | Maternal stress and infant gut microbiome | Perceived stress scale (PSS-14) Stressful life events. Unclear | 16S Phylum Genus Diversity @ 0 to 5.5 hours post delivery | No | **Maternal** race, age, marital status, Antibiotic use during pre- and perinatal periods **Infant:** Time of sampling, Mode of delivery, | Strongly cor-related taxa adjusting s in a regression model, which included maternal ethnicity/ race, age, education, marital status, time of sampling the mode of delivery and antibiotic use during pre- and perinatal period. Spearman's correlation with PRAQ only Univariate Perm nova for diversity | No statistical analysis with stress scales just observed; Actinobacteria (mean = 17%), Bacteroidetes (mean = 9.3%), Firmicutes (mean = 20%), and Proteobacteria (mean = 71%): Correlations with mothers PRAQ were analysed using Spearman correlations (This is focused on anxiety only – not extracted in this review) Pregnancy-related anxiety was significantly associated with a less diverse meconium microbiota community (p = 0.001). |

*(Continued)*

**Table 3.** (Continued)

| First Author Year Country | Aim: | Study Design Sample size = n | Variables/ Concepts in the Study PCC | Data Collection | | | | Co- variates included (not part main experiment but influence result) | Data Analysis | Key Findings: |
|---|---|---|---|---|---|---|---|---|---|---|
| | | | | Stress: Self-assessment tool: Timeline | Faecal sample test, Taxonomy Timeline | Objective Stress measure | | | Adjustment for Confounders Statistical methods | |
| Zijlmans et al. 2015 Netherlands | To prospectively investigate the development of the intestinal microbiota as a potential path-way linking maternal prenatal stress and infant health. | Longitudi-nal study n = 56 | Maternal stress and infant gut microbiome | Pregnancy related daily hassles. Perceived Stress Scale (PSS)Post Nataly Third trimester - M- 35.29 weeks (SD 1.22) | 16S Phylum Genus Diversity @ 9 time points in first 110 days | Sample of saliva cortisol | | **Maternal** age Educational level, Parity, **Infant** birthweight Breastfeeding | Breastfeeding was highlighted as a potential confounder in the analysis Multivariate analy-sis of variance Principal co-ordinate analysis (POCA) using Manhattan distances | 10% of genus level bacteria groups were significantly associated with Prenatal Daily hassles. (No p value given.) Prenatal stress recorded the greatest impact of total changes with 11% variation of the total sig different microbiota (p =0.01) Significant association with increased alpha diversity in high stress group from 7 – 80 days (No p value). infants in the high **cumulative (high report and high cortisol)** stress group tended to localize in the Escherichia, Serratia, Haemophilus Enterobacter-end of the microbiota gradient, with 853%(p <0.05) @ 28 days, 256%,1244% (p <0.01) @ 80 days, and 699% @110 days higher abundances. Infants in low **Cumulative** stress displayed higher abundance of lactobacillus, Lactococcus, aerococcus, @377% (p <0.05) 28 days, 305% (p <0.01) @ 80 days(p < 0.01) |

to investigate the association of perinatal maternal stress and the maternal gut microbiome composition, with a maternal sample only (coded with italics and bold in Table 3) [89,93]. The other five studies aimed to investigate the relationship between perinatal maternal stress and the infant gut microbiome which required a mother and infant sample [70,89,91,94,95]. Six of the studies used a cross sectional design with one a longitudinal design [90].

## Data collection

**Perinatal maternal stress.** The first objective of the review was to identify the methods and tools used to examine stress and the microbiome. Five of the studies report to examine maternal stress in the aim [70,88–91], however only two of the seven studies investigated stress as a standalone variable [70,88]. The remaining studies measured depression [91,92,95], and/or anxiety [90,91,95] along with stress, however these studies were included in this review as the data on stress could be extrapolated for results and discussion. Zijlimans [90] and Jahnke [92] were the only two studies to include post-natal maternal stress, all other studies focused on prenatal stress. There was a variety of self-assessment tools used, the most common being Cohen's perceived stress scale (PSS – 14) which was used in five of the studies [88,90,92–95]. Two of the studies used an unaltered version of the PSS – 14 tools prenatally [70,95], with Zijlmans [88] being the only study to using it postnatally. While Penalver et al. [87] used the PSS tool it was reduced to a ten-item tool. Two studies use an "Everyday problem checklist", the remaining study by Aatsinki et al [91] uses the Daily hassles negative subscale [96]. Four of the studies added these stress measurement tools to a variety of other tools to also capture anxiety, depression and pregnancy related anxiety. The most commonly used tools included The State-Trait Anxiety Inventory (STAI), Pregnancy-Related Anxiety Questionnaire-Revised (PRAQ-R), and Edinburgh post-natal scale EPDS [89,91,92,95]. Three of the studies used subjective self-assessment questionnaires as the data collections tool [70,89,95], while the other four studies also measured objective stress measurements such as hair cortisol [91], serum cytokines [88], saliva samples of cortisol [90,92]. With regards to timing of pre-natal self-assessments this ranged from 14 weeks to 41 weeks [70,88,89,94,97–99]. Post-natal time timing of assessments was 1 and 2 months [90,92].

**Gut microbiome.** The presence and abundance of bacterial taxa in the gut microbiome was tested with 16S sequencing across all the studies [69,88,93,96,99]. Data collection timepoints for the maternal gut microbiome was reported by Hechler et al [88] at a mean of 33.9 weeks, Penalver et al [91] indicates trimester 2 and 3 (week 13–41). The infant gut microbiome sampling ranged from 0 to 5.5 hours post-delivery [94], 1 month [69], 2 months [91], 2.5 months [90] to 9 timepoints in the first 110 days [89]. Alpha and beta diversity was another area investigated within the microbiome analysis in all of the studies [69,88,93,96,99].

**Taxonomic level and types.** Microbiome bacteria are often reported in taxonomic ranks ranging from less specific phylum groups to more specific species of bacteria. Across all of the studies the gut microbiome was composed of similar bacterial taxonomic groups. Common phyla included Proteobacteria [89,92,95], Actinobacteria [90,92,95], Bacteroidetes [91,92,95] Firmicutes [91,92,95], Verrucomicrobi [91], Lactic acid, Clostridia [90]. Specific genus level was identified in some studies such as Erwinia, Serratia and Haemophilus [90,91], Lactobacillus [70,89,90], Lactococcus [70,90] Pervotella timonensis [88], Akkermansia [70,92], Bifidobacterium [88,91]), Eubacterium and oscillospira [89] and clostridium [92]. This grouping of bacteria identified are important when focusing on potential implications on health.

**Perinatal maternal stress and specific taxa in the gut microbiome.** Another objective of this review was to explore the evidence on the impact of perinatal maternal stress on the bacterial taxa in the microbiome. Out of the seven studies five of the studies found that

perinatal maternal stress had a significant association with bacterial abundance in maternal gut microbiome [88] or the infant gut microbiome [70,90,91], therefore it is evident that perinatal maternal stress can alter the maternal and infant gut microbiome. Across the studies common increases or decreases in possible influential phlya and genus to health were reported in association to perinatal maternal stress However, there was some variability in these findings.

With regard to self- reported subjective questionnaires data alone, three out of seven studies identified a significant impact on bacteria taxa in the microbiome, one being a maternal gut microbiome analysis [88], while the other two were infant gut microbiome samples [69,91]. In the study by Penalver et al. (2023), PSS positively associated with species Bacteroides uniformis (p < 0.05) and Terrisporobacter mayombei (p < 0.01). Higher levels of PS also had a reported lower abundance of Akkermansia a reported probiotic (Correlation co-efficient r2 = -4). [99]. Aatinski et al. [98] indicated a positive association with maternal stress and genera Erwinia, Serratia, haemophilus (F < 0.01)(potentially pathogenic). Weiss et al [70] investigated 56 maternal and infant dyads using PSS and infant microbiome at 1 month of age. They reported a greater abundance of bacteria such as Lactobacillus (11.55 effect size (ES), p < .001), Lactococcus (8.88 (ES), p < .01) and Bifidobacterium (6.62 (ES), p < .01) in infants exposed to more stress. With significantly higher in abundance of pathogenic bacteria named as Erysipelotrichaceae (25.62(ES), p < .001),Eggerthella (23.74 (ES), p < .001), Bacteroides (22.57 (ES), p < .001), Bacteroides (22.46 (ES),p < .001), Ruminococcus (9.66 (ES), p < .01), Enterobacteriaceae (2.63 (ES), p < .05) and Enterococcus (3.66, p < .05)) for the infants whose mothers were less stressed.

Four studies combined self-report data with objective measures to assess stress, identifying new significant relationships [88,91,94,97]. Penalver et al. [88] used serum cytokine analysis to explore immune reactions along with self-reported stress. It was identified that Bacteroides negatively associated with CXCL11 (−0.02 coefficient) and positively with PSS, connecting immune factors and self-report measures. Aatinski et al. [99] reported increased hair cortisol (HCC) was related to decreased Lactobacillus (FDR < 0.01) however self-report symptoms had no associations to the HCC. One of the earliest and most well-known studies combines self-report data with maternal salivary cortisol samples to produce a cumulative stress effect [94]. This longitudinal study with 3 data collection timepoints reports a high cumulative stress group with higher abundance of pathogenic Escherichia, Serratia, Haemophilus Enterobacter-end of the microbiota gradient, with 853% (p < 0.05) at 28 days, 1244% (p < 0.01) @ 80 days, and 699% @110 days higher abundances. Infants in low cumulative stress displayed higher abundance of beneficial microbe's lactobacillus, Lactococcus, aerococcus, @377% (p < 0.05) 28 days, 305% (p < 0.01) at 80 days(p < 0.01) respectfully [94]. The daily hassles scale used identifies the greatest impact of total changes with 11% variation of the totally different microbiota (p = 0.01) [90]. Jahnke et al. [97] found significant associations with maternal and infant salivary cortisol and not with self-report of stress. Morning cortisol and cortisol awakening response (CAR) was tested for HPA axis dysregulation as a mediator of stress response. High infant basal cortisol at 3 days post-partum was associated with lower Actinobacteria (p < 0.01), Bifidobacterium (P,0.01) and higher Enterobacteriacae (p = 0.03) [97]. Low maternal CAR prepartum associated with lower abundance of Bacteroides (p = 0.01), post-partum low maternal CAR was associated higher Veillonella (p = 0.01) [97].

Two studies found no significant impact of self-reported maternal stress on the maternal gut microbiome [89], and infant gut [97]. Hechler et al. [89] found no significant associations between maternal stress measured with "everyday problem checklist and maternal gut phylum or genus level microbiota (p = 1.0). Associations were found with general anxiety and maternal microbiota at genus level (r =.71, p = 0.04) with lower anxiety recording a higher

abundance of Eubacterium and Oscillopsia (P < 0.05) Jahnke et al. also did not report any statistical significance on self-report data. Hu et al. [100] did not analyse a comparison between stress and the taxa but instead observed the abundance of the taxa in the samples and completed statistical analysis of the diversity instead, therefore has no statistical results relating to taxa.

Perinatal maternal stress on diversity of the microbiome.

The seven studies included in this review all measured diversity of the microbiome however results were inconsistent. Two of the studies measured the diversity against other scales when stress returned insignificant results, Hu et al. (2019) used general anxiety and Hechler (2019) used pregnancy related anxiety as a measured variable instead of stress therefore results are not included in this review. Two more of the studies found no significant association with perinatal stress. Aatsinki et al. [90] investigated both alpha and beta diversity and reported none of scales identified any significant associations with diversity of a 2.5 month old infant (FDR > 0.97). Similarly, Penalver et al. (2023 found no significant differences in alpha and beta diversity or votility with perceived stress scale on the maternal microbiome (no p value given).

Three studies did show an association, Ziljmans et al [91] reports a significant association with greater alpha diversity in the infant microbiome in high stress group from 7 to 80 days (no p value provided). Recently, Weiss et al. [90] recorded prenatal stress was also significantly associated with greater alpha diversity in the microbiome of a one month infant (Simpson Diversity index p = 0.02). Finally, although Jahnke et al. [70] identified a significant association between post-partum stress and diversity, this time lower alpha diversity was reported in the infant stool at 2 months of age (p = 0.01)

## Discussion

To the best of our knowledge, this is the first scoping review to map the evidence regarding perinatal maternal stress, maternal and infant gut and human milk microbiomes. Two studies were identified that researched the impact of perinatal maternal stress on the maternal gut microbiome and five studies that focused on perinatal maternal stress and the infant gut microbiome; however, no studies focused on perinatal maternal stress and the human milk microbiome. Five of the studies reported a significant association between perinatal maternal stress and the gut microbiome [67,90,93,96,97], while the remaining two studies reported no association [86,94]. The studies were undertaken in Northern Europe and the US suggesting that the research evidence is dominated by developed countries, thus indicating that gaps exist in microbiome studies from a global perspective [101]. Recent population studies show significant microbiome divergence across lifestyle, race, and ethnicity [100,102,103]. These variations are influenced by host genetics and cultural/behavioural factors like diet, hygiene, sanitation, and environment [103]. Therefore, considering race or ethnicity during data collection is crucial for microbiome analysis.

The included studies were heterogeneous in relation to stress measurement tools and assessment timepoints which may influence the variation in some of the study's findings. There is growing researcher interest on the influence of stress in health and disease, however stress is a complex variable to measure in health research [104–107]. The traditional definition of stress in psychology comes from Lazarus and Folkman's stress and coping theory, which proposes that stress occurs when individuals perceive environmental demands as exceeding their capacity to manage or adapt, depending on their supports or resilience [108,109]. Therefore, it is a multidimensional construct that comprises of exposure to events, perceptions of stress and physiological responses [106,107,110]. Research into all these domains requires multiple assessment methods/tools to investigate stress responses accurately

[104,105,111]. However, a critical barrier in measuring stress is the lack of consistency and thoroughness which requires caution when comparing data and analysing results [106,110]. The prevalence of maternal stress is reported to be up to 93% in some populations [7,9], with building evidence of infant exposure to stress during the first 1000 days linked to inherited risk of metabolic, immunologic, and neurobehavioral disorders that extend into adulthood [16,23,25,26,112,113]. This highlights the importance of reliable research to influence the development of interventions to reduce such adverse effects.

Studies in this review confirm the measurement approach taken can influence significant findings. For example, four studies that used subjective scales (Perceived stress scale, Daily hassles scale, Everyday Problem checklist) reported no significant impact of self-reported stress, however studies with additional data collected such as analysis of serum cytokines [88], salivary cortisol [90,97], cortisol awakening response [92] and hair cortisol [99] identified new significant changes to the microbiome. There are widely discussed concerns of relying on consciously perceived and self-reported stress ratings as it can introduce recall bias, lack of willingness to disclose information, with no representation of an unconscious awareness of stress [105,106,114]. Furthermore, self-report explains only a small portion of the variation in physiological stress reactivity in research [106].This limitation leads many researchers globally to incorporate similar biomarker measurements to obtain a more comprehensive and multi-dimensional approach to stress [115–117]. Despite these efforts, there is caution, as no specific biomarker for stress exists due to variability within individuals and other factors influencing hormones and cytokines [107].

In this review significant microbiome changes with self-report data often did not correlate to changes associated with objective data. This suggests that the psychological experience of stress may not always translate to measurable biological changes, and vice versa [105]. This is problematic because the link between psychological stress and poor health is believed to occur through dysregulated stress reactivity profiles [106]. Therefore, it can be hypothesised that if perinatal self-reported maternal stress is causing little to no physiological changes within the maternal gut microbiome and body, it is less likely to be transferred to an infant via neuro – immune endocrine pathways [118]. This suggests that objective measurements of stress are a necessity in this form of research investigation. Another aspect to consider when assessing stress is the duration of exposure to the stressor, and the time during the perinatal period when the stress occurs [107].

The perinatal period is most commonly described as conception to one year post-natal, with stresses reported from various sources such as emotional adjustments, relationship dynamics, financial concerns, pregnancy complications, lack of sleep, child birth problems and infant feeding issues [3,5,119]. In the studies, data collection timing varies, but three studies report data collected at multiple time points, potentially enhancing the reliability of the findings [88,97,99]. The timing of measurements is crucial to both identify the most stressful periods but also to accurately represent both chronic and acute stressors across this perinatal timeline and avoid a latency of stress exposure to measurement [80,120,121]. Additional factors also need to be considered regarding the second major concept of the review, the gut microbiome.

Defining a "healthy" infant gut microbiome is challenging due to the early life gut microbiota's dynamic changes and its responsivity to numerous, potentially confounding factors influencing its establishment [122]. For example, evidence supports the inclusion of influencing factors on the infant gut microbiome including mode of delivery, feeding mode, maternal diet, stress and antibiotic exposure [122–124]. There is some inconsistency in the adjustment of confounding factors across the studies in the review. Certain studies such as Aatinski et al. [99] include mode of delivery, feeding, antibiotic therapy and age at infant sampling

and established a positive association between the daily hassles scale and pathogenic bacteria. While in the Jahnke et al. study [97] significant associations due to stress were lost once several of the above factors were adjusted for in statistical analysis, this highlights the possibility of inflated findings if analysis is not completed accurately. Infant gut microbiome studies collected samples from 5.5 hours to 110 days, with later samples showing greater significance. Zijlmans et al. [90], the only study to look at multiple collection points also found stronger associations at 80 days old, indicating that perinatal maternal stress may have a greater impact on the microbiome after one month of age.

In the maternal gut microbiome studies, there was also inconsistency in confounding factors reported ranging from age and BMI to education, parity and civil status. A systematic review by Yang et al. [125] indicates that covariates such as pre-pregnancy and pregnancy body mass index, ethnicity, age, and co-morbidities like gestational diabetes significantly influence a mother's gut microbiota. Therefore, accurate representation of the gut microbiota should acknowledge and adjust for recognised influencers that affect composition and diversity to allow for consistent and reproducible research reporting. The results from this review identify many bacterial taxa that characterise the gut microbiome, however the significance of abundance as impacted from high and low stress varies across the studies.

Similar influential bacteria phylum and genus groups are mentioned throughout the studies that have the potential to impact future health however, some variation exists. Although some bacterial genus such as Bacteroides uniformus are reported to have dual beneficial and pathogenic roles depending if their location is in or outside the gut [126]. Some beneficial bacterial genus groups impacted by stress in this review include Bacteroides uniformus [70,88], lactobacillus [70,94,99] and Akkermansia Muciniphila [88,91]. Three of the studies identified a significant decrease of these bacteria in relation to maternal stress [88,90,91], with a study by Weiss et al [70] presenting conflicting findings of an increase of beneficial bacteria in response to stress. This may be aligned to the hypothesis by other researchers regarding some benefits of prenatal stress possibly enhancing the growth of lactobacillus in preparation to thrive in a potentially stressful postnatal environment [79].

Desirable traits for beneficial microbes include the ability to metabolise complex carbohydrates and generate short chain fatty acids, which may have positive effects on satiety and glucose metabolism and reduce the risk of metabolic disorders [120,127]. Bacteroides produce polysaccharides which have been identified to outcompete pathogens by resistance, with both Bacteroides and lactobacillus reducing autoimmune disorders by regulating the immune response and strengthening the intestinal barrier [128–130]. Furthermore, the presence of A. muciniphila has been associated with a healthy intestine and its abundance has been inversely correlated to several disease states such as remission from inflammatory bowel disease, reduced severity of appendicitis and a healthier metabolic status [131–133]. Therefore, if maternal stress reduces the abundance of these beneficial microbes, it can lead to a pro-inflammatory state in the infant's or maternal gut microbiome. This dysbiosis can recruit immune cells and disrupt metabolic processes, potentially influencing long-term health outcomes, including metabolic and immunological disorders [50,123,134].

Pathogenic bacteria groups such as proteobacteria, a substantial phylum in the human gut microbiome, includes human pathogens like Escherichia coli, Serratia, Salmonella, Helicobacter, and Erwinia is mentioned in three of the studies [70,94,99,135]. With increases reported in response to maternal stress in two studies [91,94] and a decrease reported in the study by Weiss et al [70]. Recent data suggests that these pathogens may constitute a potential microbial signature of disease, as the presence of these gram-negative bacteria is associated with sustained low-grade inflammation in the gut [136]. The abundance of Proteobacteria has been observed to increase in various inflammatory disorders, including asthma, chronic

obstructive pulmonary disease, inflammatory bowel disease, and metabolic conditions [136–138]. This may provide some evidence of the pathway to chronic disease with inherited risk from mother to baby via dysbiosis and colonisation of pathogenic microbes.

## Gaps in research

Despite emerging evidence from animal and human research over the past decade, there is a paucity of literature within these populations and concepts. For example, there was no studies focusing on maternal stress and its impact on the human milk microbiome but also none investigating maternal stress, the human milk microbiome and the infant gut microbiome simultaneously. If maternal stress is associated with alterations in the human milk micro-biome causing subsequent changes in the development of infant gut microbiome, it has the potential to provide further evidence of the source of transfer of aberrant bacteria which have been linked to various chronic diseases development.

Despite the historical belief in the sterility of human milk, recent studies have revealed a highly variable composition containing microbes, leukocytes, immune molecules, hormones, exosomes, microRNA and stems cells [34,139]. The presence of these bioactive molecules and products of metabolic activity in the milk is termed as milk metabolome and includes the entirety of metabolites present in human milk at any given time [140]. In addition to the microbiome the human milk metabolome is attracting researcher interest as it supplies both nutrients essential for infant health, along with developing microbes and prebiotic oligo-saccharides that can seed the gut microbiome of the infant [58,140]. Early evidence suggests that the human milk metabolome is also influenced by many maternal factors such as age, maternal BMI, as well as lifestyle factors such as diet and emotional health, which can in turn have a significant impact on infant health [57–59]. Despite well-established links between maternal health and human milk composition, there is a noticeable gap in studies on maternal emotional health and the human milk microbiome and metabolome, even though research in the area began approximately a decade ago [45]. Therefore, further research on the impact of maternal emotional health and specifically perinatal stress on the human milk microbiome and/or metabolome is required.

There was also limited studies with a longitudinal tracking of a possible dysbiosis or imbalance from a variable such as stress to subsequent diagnosis of a chronic health condition despite building research evidence of gut microbiome alteration to metabolic or neurodevel-opmental conditions.

## Directions for future research

The initial 1000 days of life, spanning conception to the first year of life, is a critical period where various factors can heighten the risk of adverse health conditions. It is imperative to address modifiable risk factors during this early stage to foster a "healthy" intestinal microbi-ota for mother and infant to mitigate potential risks. However, researching this area is chal-lenging due to the dynamic nature of maternal and infant gut microbiomes and the difficulty in accurately measuring stress. To avoid the underestimation or biased retrospective view of the impact of stress exposure it is important in future research to capture the psychological, behavioural and physiological responses to the exposure, along with possible contextual and individual factors that may moderate the impact of exposure and response to stress [106,107]. A robust strategy combining objective and subjective stress data, with multiple data collection timepoints for both the gut microbiome and stress, is essential to capture these variables more effectively. In addition to these factors consideration is required for influencing factors on the gut microbiome in data collection and analysis.

A primary focus is now required to explore the impact of perinatal maternal stress on the maternal gut microbiome, human milk microbiome and/or metabolome, and their connection to the seeding of the infant gut microbiome and its implications. This would involve maternal stress measurement along with, breast milk sampling and infant faecal sampling from the identified population to investigate the association of perinatal maternal stress on the breast milk microbiome and subsequently the infant gut microbiome.

Furthermore, large cohort studies are essential to survey the infant microbiome from birth through the early years, considering the maturation of the microbiome. These studies should account for inherent differences such as sex, body mass index, cultural variations, and genetics. Understanding microbial and metabolic changes, their correlation with the mentioned bacterial taxa, and their implications for long-term health outcomes, especially in inflammatory or metabolic diseases, can provide insights into the development of later pathologies.

## Strengths and limitations

This scoping review followed JBI guidance and Prisma Scr reporting guidelines to ensure robust and rigorous methodology. There were top up searches of separate concepts to ensure no research was omitted on the broad PCC question. The scoping review has some limitations that warrant consideration. The studies did differ in many ways including some researchers measuring stress in more detail than others, different confounding factors listed, with timeline difference of microbiome sampling. One study also did not complete statistical analysis of abundance and focused on diversity only and observed the taxa. Six out of the 7 studies were a cross sectional in nature, longitudinal studies can allow for a better understanding of interactions among microbial community members, as well as interactions between microbial species/genes and the human host over time [141]. Also, some studies were included that did use other measurements of stress such as anxiety and depression questionnaires if the data from stress alone could be removed. There is a concern some relevant studies could not be included where the data on stress was grouped with all questionnaires thereby possibly missing some findings in the area.

## Supporting information

**S1 File 1. "Detailed search strategy".**
(PDF)

**S1 File 2. "Prisma Scr Checklist".**
(DOCX)

**S1 File 3. "Data extraction template".**
(DOCX)

**S1 Table 1. Keywords and search.**
(DOCX)

## Acknowledgments

This scoping review was produced by author Niamh Ryan NR as a contribution to her Doctoral degree.

Contribution was made by Prof. Patricia- Leahy Warren PLW, Dr. Helen Mulcahy HM, Dr. Lloyd Philpott LP, Dr. Siobhan O 'Mahoney SOM for revision, concept analysis, screening as initialled, data curation and assistance with presentation of findings and discussion.

## Author contributions

**Conceptualization:** Niamh Ryan, Siobhain O'Mahony, Patricia Leahy-Warren, Lloyd Philpott, Helen Mulcahy.

**Data curation:** Niamh Ryan, Siobhain O'Mahony, Patricia Leahy-Warren, Lloyd Philpott, Helen Mulcahy.

**Formal analysis:** Niamh Ryan, Siobhain O'Mahony, Patricia Leahy-Warren, Lloyd Philpott, Helen Mulcahy.

**Investigation:** Niamh Ryan, Siobhain O'Mahony, Patricia Leahy-Warren, Lloyd Philpott, Helen Mulcahy.

**Methodology:** Niamh Ryan, Siobhain O'Mahony, Patricia Leahy-Warren, Lloyd Philpott, Helen Mulcahy.

**Project administration:** Niamh Ryan, Helen Mulcahy.

**Supervision:** Siobhain O'Mahony, Patricia Leahy-Warren, Lloyd Philpott, Helen Mulcahy.

**Validation:** Niamh Ryan, Siobhain O'Mahony, Patricia Leahy-Warren, Lloyd Philpott, Helen Mulcahy.

**Visualization:** Niamh Ryan, Siobhain O'Mahony, Patricia Leahy-Warren, Lloyd Philpott, Helen Mulcahy.

**Writing – original draft:** Niamh Ryan.

**Writing – review & editing:** Niamh Ryan, Siobhain O'Mahony, Patricia Leahy-Warren, Lloyd Philpott, Helen Mulcahy.

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
