## [Decision Letter · Decision Letter 0]

13 Sep 2024

PONE-D-24-25079The impact of perinatal maternal stress on the maternal and infant gut and human milk microbiomes: A scoping reviewPLOS ONE

Dear Dr. Ryan,

Thank you for submitting your manuscript to PLOS ONE. After careful consideration, we feel that it has merit but does not fully meet PLOS ONE’s publication criteria as it currently stands. Therefore, we invite you to submit a revised version of the manuscript that addresses the points raised during the review process.

We look forward to receiving your revised manuscript.

Kind regards,

Jayakrishna Madabushi, MD

Academic Editor

PLOS ONE

Journal Requirements:

1. When submitting your revision, we need you to address these additional requirements. Please ensure that your manuscript meets PLOS ONE's style requirements, including those for file naming. The PLOS ONE style templates can be found at https://journals.plos.org/plosone/s/file?id=wjVg/PLOSOne_formatting_sample_main_body.pdf and https://journals.plos.org/plosone/s/file?id=ba62/PLOSOne_formatting_sample_title_authors_affiliations.pdf 2. Please amend the order of the figures, as the PRISMA flowchart has to be Figure 1 for systematic reviews. You may move the other figure to later in the manuscript or to the supplemental information. 3. We note that your Data Availability Statement is currently as follows: "All relevant data are contained within the manuscript and/or supporting information files." Please confirm at this time whether or not your submission contains all raw data required to replicate the results of your study. Authors must share the “minimal data set” for their submission. PLOS defines the minimal data set to consist of the data required to replicate all study findings reported in the article, as well as related metadata and methods (https://journals.plos.org/plosone/s/data-availability#loc-minimal-data-set-definition). For example, authors should submit the following data: - The values behind the means, standard deviations and other measures reported;- The values used to build graphs;- The points extracted from images for analysis. Authors do not need to submit their entire data set if only a portion of the data was used in the reported study. If your submission does not contain these data, please either upload them as Supporting Information files or deposit them to a stable, public repository and provide us with the relevant URLs, DOIs, or accession numbers. For a list of recommended repositories, please see https://journals.plos.org/plosone/s/recommended-repositories. If there are ethical or legal restrictions on sharing a de-identified data set, please explain them in detail (e.g., data contain potentially sensitive information, data are owned by a third-party organization, etc.) and who has imposed them (e.g., an ethics committee). Please also provide contact information for a data access committee, ethics committee, or other institutional body to which data requests may be sent. If data are owned by a third party, please indicate how others may request data access.

Reviewers' comments:

Reviewer's Responses to Questions

**Comments to the Author**

1. Is the manuscript technically sound, and do the data support the conclusions?

Reviewer #1: Yes

Reviewer #2: Yes

2. Has the statistical analysis been performed appropriately and rigorously? 

Reviewer #1: Yes

Reviewer #2: Yes

3. Have the authors made all data underlying the findings in their manuscript fully available?

Reviewer #1: Yes

Reviewer #2: Yes

4. Is the manuscript presented in an intelligible fashion and written in standard English?

Reviewer #1: Yes

Reviewer #2: Yes

5. Review Comments to the Author

Reviewer #1: I've read through the manuscript, and I find it to be technically sound, with the data presented largely supporting the conclusions drawn. The scoping review method you've chosen is well-suited to the research question, and it's clear that you've adhered to established guidelines, such as the Joanna Briggs Institute’s methodology and PRISMA-ScR reporting standards.

The statistical analyses seem to be handled appropriately, and the rigor is evident. That said, I did notice some significant heterogeneity among the included studies, which could impact the generalizability of your findings. It might be helpful to delve a bit deeper into how this heterogeneity could influence the overall conclusions.

I think the discussion could benefit from a bit more depth, particularly when it comes to the methodological challenges and inconsistencies across the studies. Addressing these issues more thoroughly could provide readers with a better understanding of the current state of research in this area and offer clearer guidance for future studies.

Reviewer #2: Thank you for the opportunity to review this manuscript. The study does a great job exploring how perinatal maternal stress affects the maternal and infant gut and human milk microbiomes. The manuscript is clear and well-organized. However, there are a few areas where adding more detail and clarity could further strengthen the overall study.

Introduction: Consider addressing the impact of maternal nutrition on maternal and infant microbiomes during the perinatal period. Maternal diet is a critical factor that can influence the composition and function of the microbiome, potentially affecting maternal health, pregnancy outcomes, and infant development.

Methods : Please include a discussion on the quality assessment of the included studies.

Consider addressing potential biases such as selection bias, publication bias, and reporting bias, and their possible influence on the review’s results.

Results and discussion : Explain how the findings might be applied in clinical settings. Consider how the findings of your review might influence guidelines for prenatal care, particularly regarding stress management

Indicate whether a sensitivity analysis was considered or conducted to evaluate the robustness of the findings.

Consider addressing the variability in how studies define types of stress (e.g., psychological vs. physical, acute vs. chronic) and the tools used to measure stress (e.g., self-reports, physiological measures).

Discuss the potential confounding effects of comorbid conditions (e.g., depression, anxiety, or physical health issues) that could influence both maternal stress and microbiome composition.

6. PLOS authors have the option to publish the peer review history of their article (what does this mean? ). If published, this will include your full peer review and any attached files.

**Do you want your identity to be public for this peer review?** For information about this choice, including consent withdrawal, please see our Privacy Policy .

Reviewer #1: No

Reviewer #2: **Yes: ** Mohsin Raza

---

## [Author Response · Author response to Decision Letter 0]

4 Oct 2024

Please see Response to reviewers letter attached.

---

## [Decision Letter · Decision Letter 1]

14 Jan 2025

The impact of perinatal maternal stress on the maternal and infant gut and human milk microbiomes: A scoping review

PONE-D-24-25079R1

Dear Dr. Ryan,

We’re pleased to inform you that your manuscript has been judged scientifically suitable for publication and will be formally accepted for publication once it meets all outstanding technical requirements.

Kind regards,

Juan J Loor

Academic Editor

PLOS ONE

Additional Editor Comments (optional):

Reviewers' comments:

Reviewer's Responses to Questions

**Comments to the Author**

1. If the authors have adequately addressed your comments raised in a previous round of review and you feel that this manuscript is now acceptable for publication, you may indicate that here to bypass the “Comments to the Author” section, enter your conflict of interest statement in the “Confidential to Editor” section, and submit your "Accept" recommendation.

Reviewer #2: All comments have been addressed

2. Is the manuscript technically sound, and do the data support the conclusions?

Reviewer #2: Yes

3. Has the statistical analysis been performed appropriately and rigorously? 

Reviewer #2: N/A

4. Have the authors made all data underlying the findings in their manuscript fully available?

Reviewer #2: Yes

5. Is the manuscript presented in an intelligible fashion and written in standard English?

Reviewer #2: Yes

6. Review Comments to the Author

Reviewer #2: Please elaborate on which dietary factors or specific nutrients are shown to influence the maternal and infant microbiome.

Everything else has been addressed adequately.

7. PLOS authors have the option to publish the peer review history of their article (what does this mean? ). If published, this will include your full peer review and any attached files.

**Do you want your identity to be public for this peer review?** For information about this choice, including consent withdrawal, please see our Privacy Policy .

Reviewer #2: **Yes: ** Mohsin Raza

---

## [Editor Report · Acceptance letter]

PONE-D-24-25079R1

PLOS ONE

Dear Dr. Ryan,

I'm pleased to inform you that your manuscript has been deemed suitable for publication in PLOS ONE. Congratulations! Your manuscript is now being handed over to our production team.

Kind regards,

on behalf of

Dr. Juan J Loor

Academic Editor

PLOS ONE